# Long-Term Loss of Coral Reef in the Gulf of Aqaba Estimated from Historical Aerial Images

**Elad Topel and Yoav Lehahn ***

Department of Marine Geosciences, Charney School of Marine Sciences, University of Haifa,
Abba Khoushy Ave 199, Haifa 3498838, Israel
* Correspondence: ylehahn@univ.haifa.ac.il

**Abstract:** Located at the northern tip of the Red Sea, the Gulf of Aqaba coral reef is highly susceptible to anthropogenic pressure associated with the massive development of the two neighboring cities of Eilat and Aqaba. Over the years, the reef has been extensively studied in a number of research and monitoring programs, which provide detailed information on a variety of ecological, physiological, biogeochemical and physical variables. Due to the local nature of these surveys, although the state of the reef at specific times and locations is very well characterized, long-term changes in the spatial characteristics of the reef as a whole have not yet been quantified. Here, through analysis of historical aerial images, we address one of the most important aspects of this knowledge gap and quantify the decrease in coral reef coverage, with respect to a baseline level prior to the establishment of the City of Eilat in 1949. Our results show that along a 8460 m segment of the shoreline at the northwestern edge of the GOA (29°29′–30°N/34°54′–35°E), infrastructure construction has led to the loss of more than 4810 m$^2$ of coral reef. This loss has been caused either directly by the construction (most importantly the Eilat Port) or indirectly by changing environmental conditions, which in turn damage the reef. Our results show that historical aerial images can provide a unique source of information on the spatial characteristics of marine and coastal systems prior to the era of Earth observation satellites.

**Keywords:** Gulf of Aqaba; coral reef; historical aerial images; long term monitoring; coastal infrastructures

## 1. Introduction

Coral reefs are among the most biologically rich and productive ecosystems on earth [1–3]. They provide valuable ecosystem services to millions of people living in coastal communities, being an important source of food and income, serving as nurseries for commercial fish species, attracting divers and snorkelers, promoting tourism, and protecting shorelines from erosion due to storms [1,4,5]. Coral reefs are susceptible to various stressors, induced directly or indirectly by human activity [6,7]. Reef-building corals around the world are going through degradation and bleaching due to ocean warming and acidification, associated with climate change [4,8,9], and overall reefs are declining worldwide [1,5,7]. The anthropogenic impact on coral reefs is most pronounced in the vicinity of coastal cities, which have a substantial influence on the marine and coastal environment [10]. Recent studies examining the influence of urban development on coral reefs show that over 60% of the world's reefs experience damage from local activities such as overfishing, coastal development, and watershed pollution [7,10]. Urban development also affects nearby reefs by reducing light penetration, elevating plankton productivity favoring eutrophication, covering reefs with piers and piling, changing shoreline morphology through enhancement of sand accumulation, and more [11,12].

The Gulf of Aqaba (GOA), at the northern tip of the Red Sea, hosts one of the northernmost tropical reefs in the world. The GOA coral reef has an exceptionally high bleaching resilience [10] but is highly susceptible to anthropogenic pressure associated with the massive development of the two neighboring cities of Eilat and Aqaba (Figure 1). The GOA coral reef, and especially its northwestern edge, has been extensively monitored

and studied since the 1960s, resulting in a very large body of knowledge on different aspects the reef's ecosystem e.g., [2,10,13–15]. As in other parts of the world e.g., [5,16], the GOA coral reef monitoring efforts are mostly based on in situ surveys, in which variables such as reef community structure, mortality rate, and cover are measured at a very high resolution across transects e.g., [14]. While such surveys provide detailed information on a large number of key ecological variables, due to logistic complications and high costs they are very limited in their spatial and temporal coverage. As a result, while the state of the reef at specific times and locations is very well characterized, long-term changes in the spatial characteristics of the reef as a whole have not been quantified. This limitation can be partly overcome by using information from cameras and sensors aboard drones, airplanes, and satellites [3,16–23] These remote sensing data allow mapping changes in various reefs' properties over multiple temporal (from days to decades) and spatial (from cm to km) scales, providing important complementary information to that collected through in situ surveys.

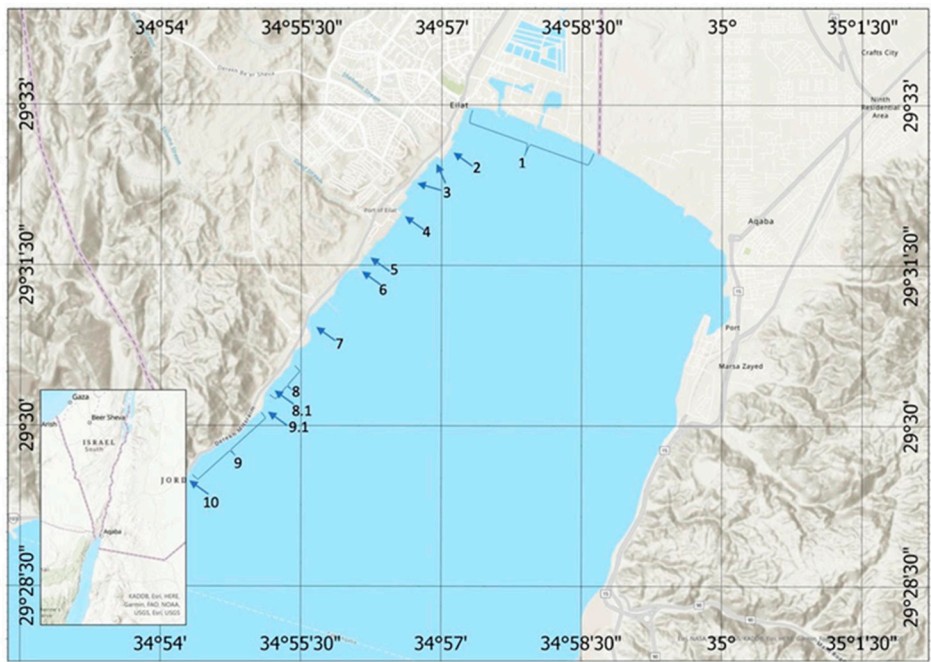

**Figure 1.** The Gulf of Eilat, located at the northern tip of the Red Sea (see insert). The numbers indicate subregions discussed in the text: 1. North Shores, 2. Navi, 3. Hachashmal and Hadekel Beach, 4. Port of Eilat, 5. Dolphin Reef, 6. Pier 1 (Ecological Beach), 7. South Marina, 8. The Coral Beach reserve (Closed Reserve), 8.1. The Japanese Gardens, 9. The South Shore reserve (Open Reserve) and Sharem Al' Sheikh Rd., 9.1. The Migdalor Beach, 10. International border.

In this work, we harness remote sensing data to characterize spatial changes in the GOA coral reef associated with the accelerated development of the nearby city of Eilat, which was founded in 1949. Specifically, through acquisition and analysis of a series of historical aerial images that go back to 1945, we quantify how reef cover has changed over time along a 8460 m segment of the shoreline at the northwestern edge of the GOA, with respect to a baseline coverage prior to the city of Eilat's construction.

## 2. Materials and Methods

### 2.1. Aerial Images

The study is based on an analysis of a series of historical and current-day aerial images taken over the northwestern edge of the GOA, off the coast of Israel, between the years 1945 (prior to the construction of the city of Eilat) and 2015 (Table 1). Browsing through the archive of the Survey of Israel (https://www.mapi.gov.il/en/Pages/default.aspx accessed on 19 September 2022), we found a series of six images, roughly one set per decade, in which

the reef was detectable along the gulf's Israeli shoreline (Figure 1). The five earlier images (1945, 1957, 1969, 1977, and 1995) are composed of scanned black-and-white photos with varying spatial resolution, while the most recent image (2015) is a geometrically corrected (orthophoto) colored image (Table 1). This historical dataset is complemented by current-day drone data with a spatial resolution of 1 cm/pixel, which are used for classification and validation at specific locations. In addition to the historical aerial images, we have obtained high resolution data in specific locations, using a Mavic Pro2 drone equipped with 1″ CMOS sensor, 20 million effective pixels, field of view (FOV) of 77 degrees, 35 mm format equivalent length to 28mm, aperture range f/2.8–f/11 and shooting range from 1m and on [24].

**Table 1.** Aerial images used in this research.

| Year | Spatial Resolution (cm/pixel) | Number of Images Composing the Series | Type | Flight Date |
|---|---|---|---|---|
| 1945 | 1000 cm/pixel | 1 | Scanned Black and White Aerial photograph | May 1945 |
| 1957 | 200 cm/pixel | 6 | Scanned black and white aerial photograph | April 1957 |
| 1969 | 50 cm/pixel | 12 | Scanned black and white aerial photograph | April 1969 |
| 1977 | 50 cm/pixel | 5 | Scanned black and white aerial photograph | January 1977 |
| 1995 | 50 cm/pixel | 9 | Scanned black and white aerial photograph | April 1995 |
| 2015 | 25 cm/pixel | 1 | Orthophoto | April 2015 |
| 2019 | 10 cm/pixel | 9 | Drone Images | July 2019 |

*2.2. Image Processing and Analysis*

Image processing and analysis was performed with ESRI's (https://www.esri.com accessed on 19 September 2022) ArcGis-Pro 2.8 (www.arcgis.com accessed on 19 September 2022). At a first stage, all images were rectified with respect to the 2015 orthophoto using the Georeference tool. The rectification was performed based on colocalization of at least 12 clearly identified points located on land or on marine infrastructures. After image rectification, a two-step classification process was performed. In each image, an unsupervised classification was performed using the Iso Cluster Unsupervised Classification tool to distinguish between water and land. The border between the two regions was defined as the shoreline. Accordingly, long-term changes in shoreline morphology were quantified by comparing the lines delineated on the 1945 and 2015 images [25–28]. Characterization of coral reef in the images was done through unsupervised classification using the Iso Cluster Unsupervised Classification tool using 6 classes. The classification was done over a limited number of regions of interest (ROI), with an area according to the area of interest. In each ROI, 6 classes were identified, corresponding to the distinctly identified features of 1. deep water; 2. shallow water; 3. deep reef; 4. shallow reef; 5. sand; and 6. terrestrial features. The shallow and deep reefs were grouped into a single cluster defined as a detectable reef whose surface area, as extracted from the images, was used as an estimated reef surface area (ERSA) at a given time.

**3. Results and Discussion**

Quantification of changes in coral reef coverage from a series of historical aerial images requires reliable delineation of the reef boundaries. To estimate the reliability and consistency of our approach, we first compared the spatial patterns of the coral reef, as emerged from analysis of the different aerial images within a region that is characterized by little spatiotemporal change (Figure 2). Specifically, we focused on the area of the Eilat Coral Beach Nature Reserve (https://en.parks.org.il/reserve-park/eilat-coral-beach-nature-reserve-2/ accessed on 19 September 2022), which is operated by the Israel Nature and Parks Authority (https://en.parks.org.il/ accessed on 19 September 2022).

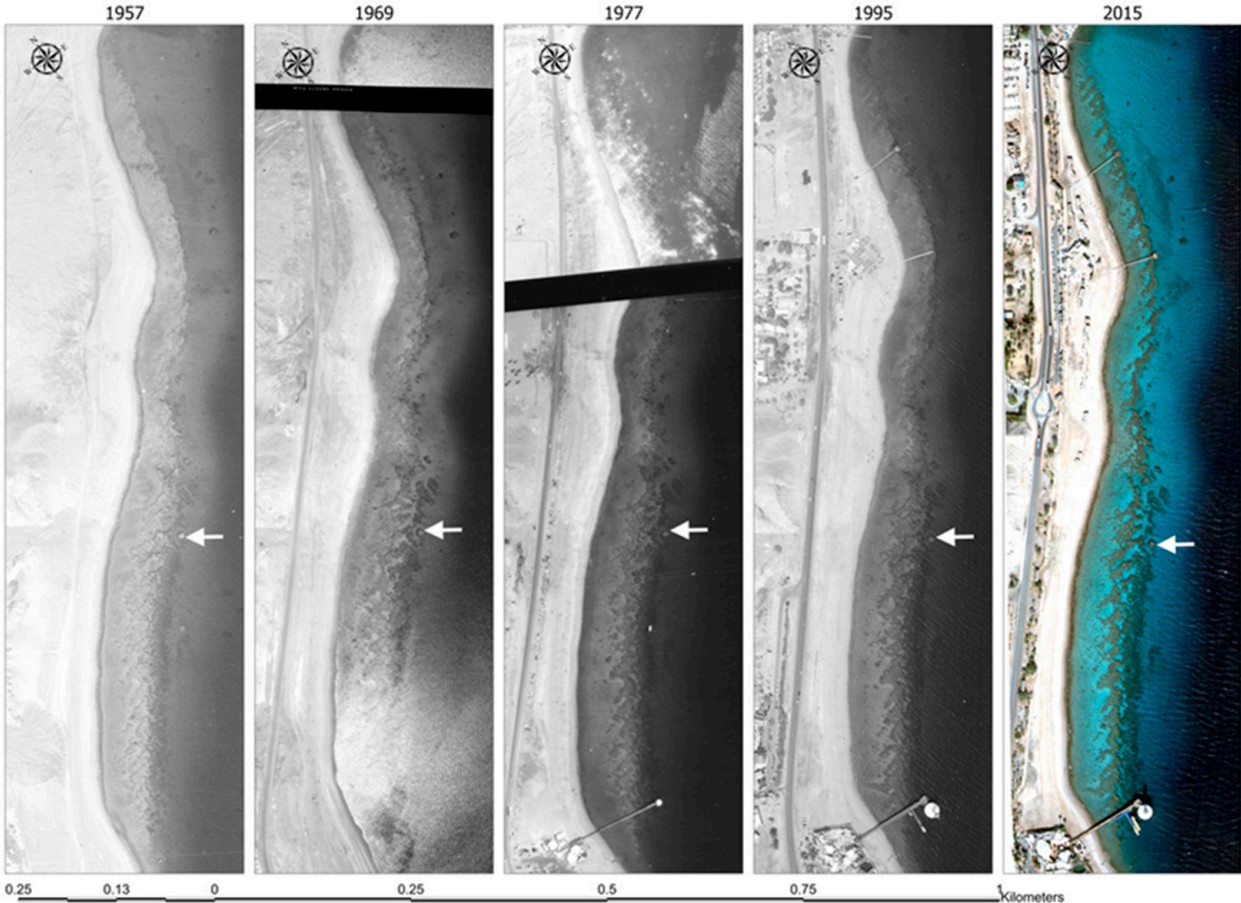

**Figure 2.** Historical aerial images over the area of the Eilat Coral Beach Reef Nature Reserve (the closed reserve, region 8 in Figure 1) between the years 1957 and 2015, emphasizing the similarity in the reef's spatial characteristics. White arrow points to the area shown in Figure 3.

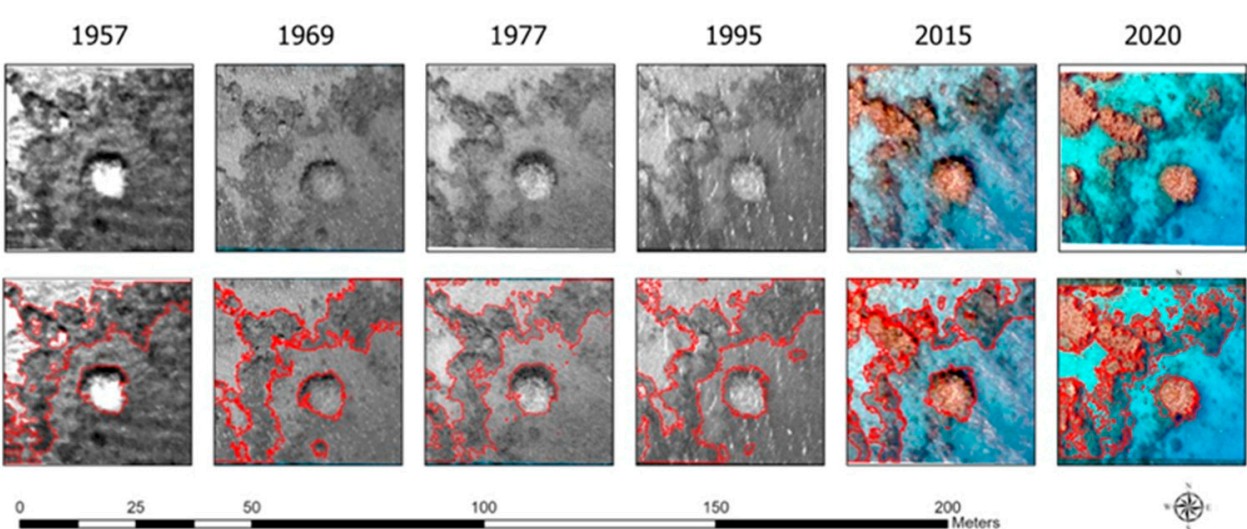

**Figure 3.** A segment of the coral reef located within the reserve (location marked by white arrow in Figure 2), as observed in historical aerial images taken between the years 1957 and 2015, and in a drone-based image taken in 2020. Red contour in the lower panels delineates the border of the reef, as identified in each image.

The reserve stretches over a 1.2 km long segment of the coral reef, approximately 5.5 km from Eilat city center (see Figure 1 for location). The nature reserve is partitioned into a closed reserve (the South Shore reserve, region 8 in Figure 1) and an open reserve (the Coral Beach reserve, region 9 in Figure 1), which were established in 1964 and 2009, respectively. Here, we focus on the older closed reserve. Within this region, reef classification yields very similar results when applied on the different images analyzed (Figure 2). Comparison between the 1957, 1995, and 2015 images, which were not interrupted by clouds and sun glint, shows a maximum difference of approximately 10% in ERSA. This difference is attributed to variations in image quality and light conditions. The reliability of our approach was further validated by focusing on a distinct feature located at the southern part of the reserve (Figure 3). The similarity in reef morphology and ERSA (maximum change of 15%) as observed in the different images (1957, 1969, 1977 and 1995 in Figure 3) and in a high-resolution drone image of the area (2015 in Figure 3) demonstrate the ability of our approach to delineate reef area in an accurate and consistent manner. In addition to emphasizing the accuracy of the reef classification process, this analysis provides an example for a segment of the reef whose physical structure was not significantly changed over the past seven decades, possibly pointing to a preservation success of the Nature and Parks Authority.

Our analysis of human-induced changes in coral reef coverage is focused on the effect of marine and coastal infrastructures. We distinguished between two types of effects: direct effect of reef elimination by construction and indirect effect where the infrastructure and/or its construction process changed the environmental conditions, which in turn damage the reef. Examples of these two effects are described below.

The most remarkable example for direct human impact on the reef's physical structure is the construction of the Port of Eilat (Figure 4). The port was constructed in its current location during the beginning of the 1960s and has been operational ever since. At the time of its inauguration in 1965, the port covered a total area of 121,000 m$^2$. During the 1970s, the port was expanded northward, adding approximately 12,500 m$^2$ to its area. The construction of the port in the 1960s and its expansion in the 1970s led to the direct elimination of 3355 m$^2$ and 500 m$^2$ of coral reef, respectively (Figure 4). After the expansion of the port in the 1970s, no evident changes in the reef's coverage were observed.

The indirect effect of marine infrastructure on the coral reef is exemplified for the case of Pier One, which is located approximately 3.5 km south of Eilat city center (Figure 5). The pier was constructed in 1969 as part of a national-scale pipeline operated by the Europe Asia Pipeline Co., Ltd. (https://www.eapc.com/ accessed on 19 September 2022). The pipeline is used for transporting oil between the Gulf of Aqaba and the southeastern Mediterranean. The pier is a concrete-and-stone infrastructure penetrating 150 m into the sea, which affects the reef both directly and indirectly. The pier's construction led to a direct loss of 255 m$^2$ of coral reef (Figure 6). In addition, since its construction the pier has been indirectly affecting the adjacent reef through blocking the alongshore sand transport, leading to constant accumulation of sand at an annual rate of 75 m$^2$ year$^{-1}$. The area covered by the accumulated sand has been increasing at an annual rate of 15 m$^2$ year$^{-1}$, resulting in an overall coral reef loss of 955 m$^2$ between the years 1957 and 2015.

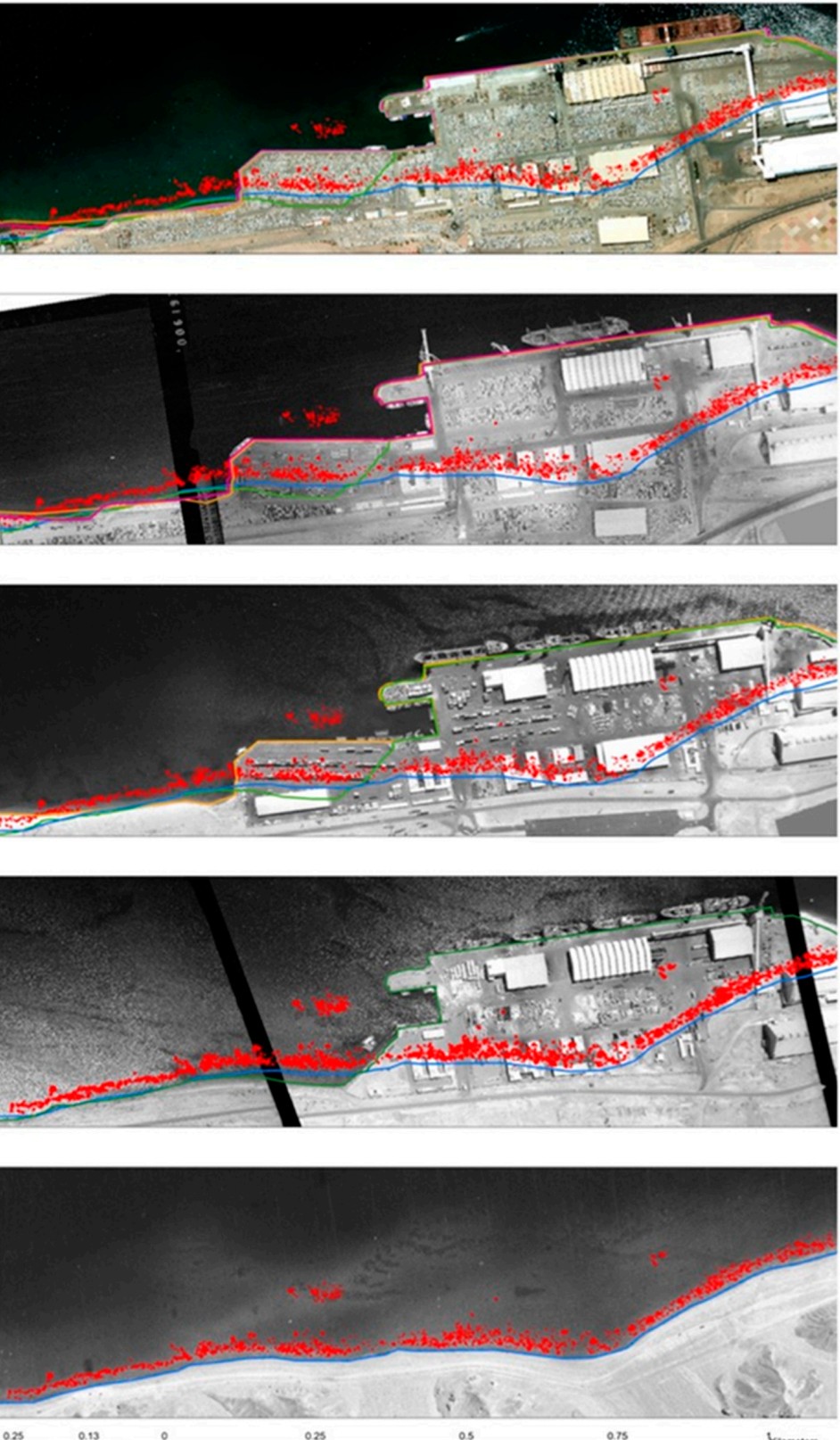

**Figure 4.** Historical aerial images over the area of Eilat Commercial Port (region 4 in Figure 1) between the years 1957 and 2015. Red contour delineates the border of the reef as identified in the 1957 image. Blue, Green, Brown, Purple contours delineate the shoreline as identified in the 1957, 1969, 1977, 1995, and 2015 images, respectively.

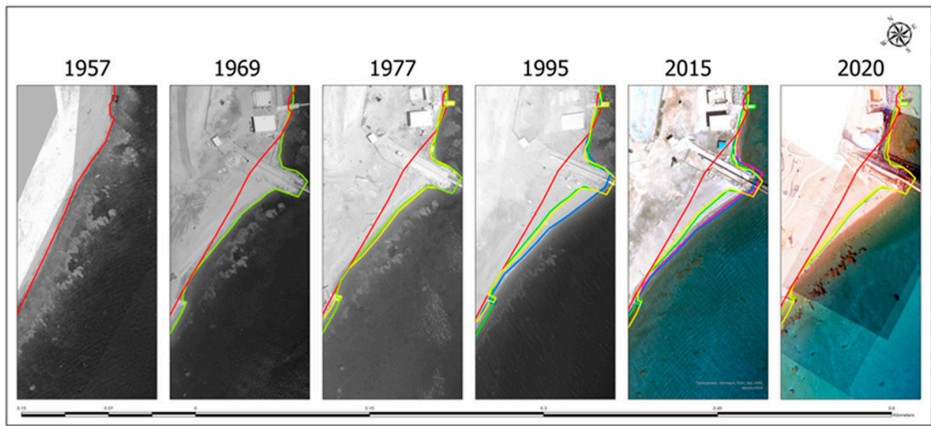

**Figure 5.** Historical serial images from the years 1957–2015, and a drone image from 2020, over the area of Pier One (region 6 in Figure 1). Red, Green, Yellow, Blue, and Purple lines delineate the shoreline as identified in the 1957, 1969, 1977, 1995, and 2015 images, respectively.

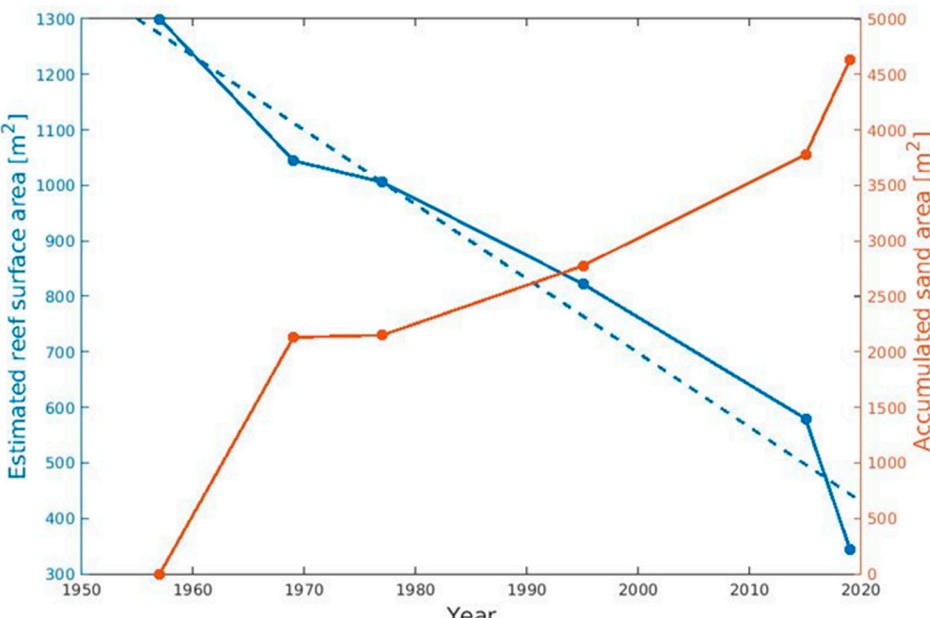

**Figure 6.** Changes in the estimated reef surface area (ERSA, solid blue line) and accumulated sand area (orange line), in the vicinity of Pier One between the years 1957 and 2015. Dashed line shows the linear trend in ERSA ($R^2 = 0.96$).

## 4. Summary and Conclusions

Using a set of historical aerial images, we provided a unique perspective on changes in the spatial distribution of coral reefs along an 8460 m segment of the shoreline in the northwestern edge of the Gulf of Aqaba during the past seven decades and related them to human activity. This extended set of observations goes back to the period prior to the construction of the adjacent city of Eilat, providing a baseline reference for the spatial characteristics of the reef before it was substantially influenced by the intense human activity associated with the city's development.

Most of the coral reef loss documented in this work is associated with expansion of the shoreline (defined as the interface between water and land) due to construction of infrastructures. Between the years 1957 and 2015, this expansion covered a total area of 184,000 m² (Figure 7), leading to a loss of more than 4810 m² of coral reef, mostly in the southern part of the coast. The main cause for the observed coral reef loss is the construction of infrastructures in the sea, most importantly the Eilat port and Pier One.

However, the reef was also substantially affected by infrastructure construction on land. This is well-exemplified in the open reserve area, where expansion of the southern road towards Sharem Al' Sheikh in 1971 changed the slope of the shore, leading to accumulation of sand over the lagoon. In contrast, one of the areas in which we did not observe any change in shoreline morphology and in reef coverage is that of the closed reserve. Although preservation activities in the closed reserve do not affect construction on land, the fact that we did not observe any change in reef area there may point to a preservation success of the Nature and Parks Authority.

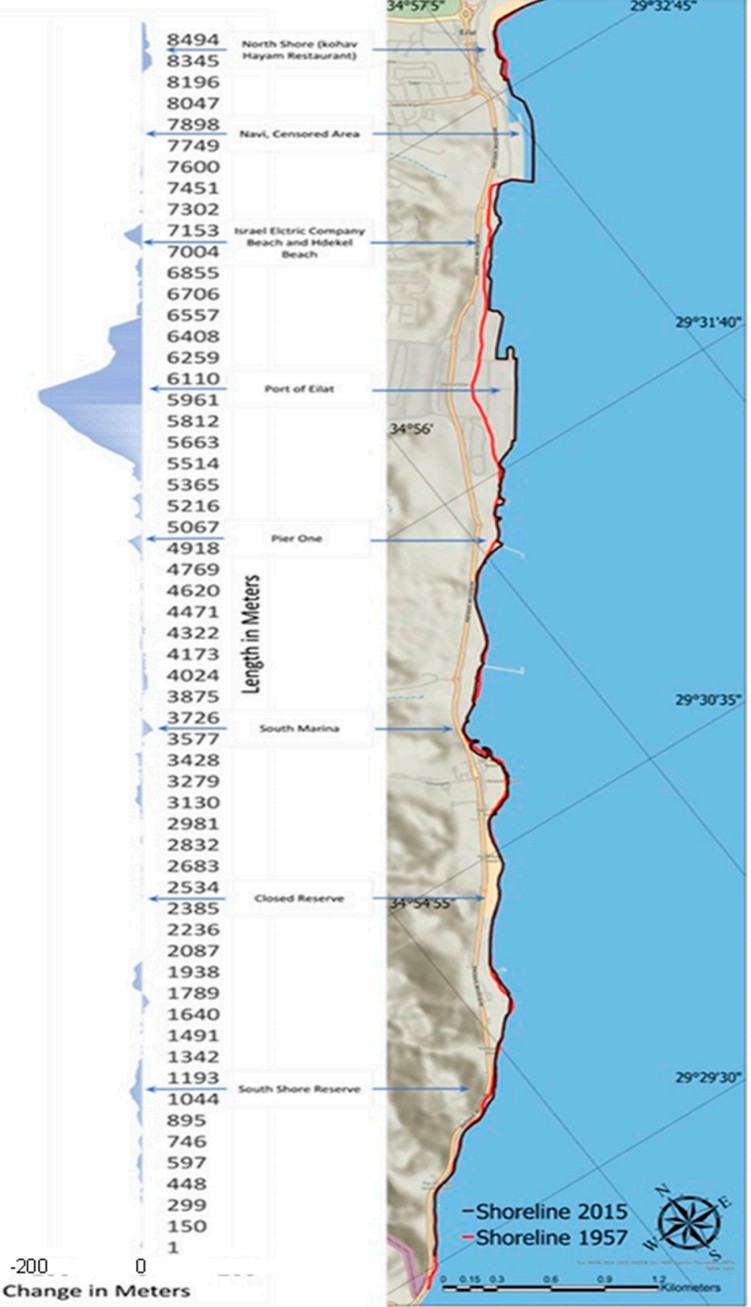

**Figure 7.** Changes in the morphology of the Israeli shoreline of the Gulf of Aqaba over six decades. Right panel: the shoreline, defined as the border between water and land, as delineated based on the 1957 (red line) and 2015 (black line) images. Left panel: distance, in meters, between the two shorelines, with negative values indicating expansion of land towards the sea.

In general, reef loss can result directly from infrastructure construction or indirectly from processes that are triggered or intensified by the construction. Unlike direct effects, which are instantaneous in nature, indirect consequences of infrastructure construction, such as accumulation of transported sand over the reef, may be associated with prolonged processes whose damage increases with time (e.g., the Pier One area). Therefore, although being responsible for a relatively small part of the overall coral reef loss, indirect consequences of human activity call for operational measures that will reduce future damage.

While our analysis, which largely relies on information from black-and-white aerial images, provides a reliable quantification of long-term changes in reef coverage, it does not provide any information on ecological or physiological changes in the state of the reef. Moreover, the spatial resolution of our analysis does not allow identification of small-scale changes associated with activities such as diving tourism and fishery, which may also damage the reef. Remote-sensing data, including products not used here as chlorophyll concentrations and turbidity, should therefore be used in combination with in situ measurements from long-term research and monitoring programs e.g., [2,10,13–15]. In addition, data collection and analysis should be extended to areas not covered here, in order to track changes in the GOA coral reef as a whole.

Historical aerial images provide a unique source of information on the spatial characteristics of marine, coastal, and terrestrial systems prior to the era of Earth observation satellites. The results presented here open the way to the utilization of such images for quantifying very long-term changes in the spatial characteristics of coral reefs, providing a unique perspective on the spatial characteristics of coral reefs prior to the influence of intense human activity.

**Author Contributions:** E.T. and Y.L. conceived the research and wrote the paper. E.T. processed and analyzed the data. Y.L. oversaw the research. All authors have read and agreed to the published version of the manuscript.

**Funding:** This research received no external funding.

**Data Availability Statement:** Data was obtained from the Survey of Israel and are available with their permission.

**Acknowledgments:** We thank the Survey of Israel for providing the aerial images used in this research.

**Conflicts of Interest:** The authors declare no conflict of interest.

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
