# Peer review of "Long-Term Loss of Coral Reef in the Gulf of Aqaba Estimated from Historical Aerial Images"

_remotesensing, doi:10.3390/rs14246305_

Round 1

Reviewer 1 Report

Dear authors,

I commend you for digging out a treasure trove of historical aerial photos over the GOA reef area and making an effort to estimate the long-term reef cover losses. When this type of data is available, it's good to make use of them. I have put together my comments on the manuscript in a word document attached. Unfortunately there is an unknown amount of text that has gone missing from line 250 and forward. I can therefore not recommend the editor to accept the manuscript for publishing directly, but after addition of the missing text and minor revisions this manuscript should be a good addition to the knowledge base on the reef of the Gulf of Aquaba. 
Yours sincerely

Reviewer 2 Report

The MS reports some important results giving evidence to the effects of building the Eilat port and the following urbanization on the local coral reef. I raccomand  to connect, as much as possible, these resuts with all those concerning the other areas of the Akaba Gulf or, if not possible to the lack of data, to underline the need to collect such data in order to have a wider view of the gulf as a whole. Overall a wider connection of these original and meaningful data with the hydrodynamics and with the bio-ecological knowledge of the Gulf of Akaba can also improve this paper.

Some points  that must be re-addressed are reported in the following.

ABSTRACT

line 7 Please, add here the geographic coordinates of the area.

      9 "the " - area (?)

      11 "..due.. while..  " - delete while, change in: ..however, long-term changes in the reef as a whole....

      17 "...4810 m2 - please add here the percentage: equal to x % of the estimated local reef area..

 INTRODUCTION

line 26 ".. these diverse systems provide.." - they provide

36 " light availability.." -Light penetration

" elevating plancton production .." - favouring eutrophication

41 " natural selection.." - We cannot exclude natural selection occurred, howeverbut the depts the GOA (1850 mt!) and the neigbour Red Sea reach, as well as local upwelling may be the cause of such resistence of corals. Plese add here something about the local termperature series and, if they are lacking, underline the need of such data...

46 - Please, cite here papers and data on the Eastern edge of the Gulf of Akaba and other areas and, if they are laking, underline this gap of knowledge and research in the whole Akaba Gulf area..

65-66 "...the obtained results .... " All this sentence is not necessary, please, delete.

Fig. 1 - Add the geographic coordinates (Lat-Long) also to the figure.

M&M

LINE 81 " The study is based on the analysis of a series of historical..." - This study is based on  a series of ..

136 " ..defined as a detectabe reef " - Change in:  - defined: "detacteble reef"

RESULTS AND DISCUSSION

line 226 "..significant.." - If there is not any statistics, " evident" should be better..

Fig.  5 - Could you evidence with the red color  coral reefs in 1957 and in the following years?

CONCLUSIONS

Pag 282  "..spatial characteristic of the the Gulf of Akaba coral reefs"...        -change in : " the spatial distribution of coral reefs in the Israelian north-western edge of the Gulf of Akaba..

290 -Please, add a percentage (%) here too.

311 - add here: based also on clorophyl remote sensing monitoring.(Clorophyl monitoring may help to evaluate the ealth of shallow coral reefs, primary productivity and  eutrophication of the area). Also the study of gulf hydrodynamics could help our understanding of the Gulf of Akaba ecosystem.

At the end of the paper You shoudl also add a sentence like:  Long-term remote sensing monitoring, as well as all the research on coral reefs should be also extended to the whole Gulf of Akaba.....

Fig. 7 Add the geog. coord. to the figure.

Reviewer 3 Report

The manuscript is a mere description of different photos comparisons from 1945 to 2019, just to conclude that the reduction of coral reef area was mainly due to the construction of Port of Eilat. Perhaps comparing an old map with an actual one would lead to the same conclusion. There is no detailed analysis on the changes of the remaining coral area and of its health over time. The final Figure (Figure 7), which could have been a resume of all the previous ones, only shows the changes (in meters) of the coastline. But are those changes positive or negative? In some areas there was a loss of mainland, while in others there was a gain (with a loss of the aquatic ecosystem and of the corals). Can the effect of sea level rise be detected in those changes? Where, along that coastline, could changes of coral reef be detected (not only in the constructed areas)?

I do not agree with the final conclusions:” The results presented here open the way to the utilization of such images for quantifying very long-term changes in the spatial characteristics of coral reefs, providing a unique perspective on the state of coral reefs prior to the influence of intense human activity.”

The authors were not able to show any long-term changes on the state of the corals except the coral area loss due to land reclamation and harbour construction. There is no analysis on the “state” of coral reefs.

Some other comments:

Figure 2. I do not understand what we are supposed to see. The red in the figures is the coral in 1957. Which are the differences within the several images, apart from land reclamation due to harbour construction? Was the coral distribution exactly the same over time? Perhaps not! A detailed, and much more interesting analysis is missing.

The same for Figure 4.

Round 2

Reviewer 3 Report

The manuscript has improved. However, I still think that the subject has reduced interest since is a mere observation of sequential images just to show the impact of the construction of Eilat port.

If published some aspects need to be improved:

Abstract - The "loss of 4810 m2 of coral reef" corresponds to which percentage of the total coral area studied? What is / was the total coral area?

Figures 4 and 5. The scale of the maps need to be correctly indicated, using the same procedure as in the previous maps, where a given distance in the map corresponds to a real distance of XXX km (or m).

Figure 7: If only positive changes were observed - land gain, the red line should always be before the black line. That does not happen in several parts of the map.  Examples: northern part of the map, in the middle, ...
